# CURRICULUM DYNAMIC GRAPH INVARIANT LEARNING UNDER DISTRIBUTION SHIFT

## ABSTRACT

Dynamic graph neural networks have attracted intensive research interests recently but generally suffer from handling distribution shifts that widely exist in dynamic graphs. Although the existing works attempt to disentangle the invariant and variant patterns, they ignore the training status of the graph neural network and the importance of training samples at different times, which are critical to model invariant patterns accurately in dynamic graphs under distribution shifts. In this paper, we study distribution shifts in dynamic graphs with curriculum learning for the first time, which remains unexplored and faces the following challenges: (i) how to design a tailored training status evaluation strategy; and (ii) how to design a tailored sample importance reweighting strategy, so as to handle distribution shifts in dynamic graphs. To address these challenges, we propose a Curriculum Dynamic Graph Invariant Learning (**CDGIL**) model, which can handle distribution shifts in dynamic graphs by capturing and utilizing invariant and variant patterns guided by the proposed curriculum learning strategy. Specifically, we first propose a dual disentangled dynamic attention network to capture the invariant and variant patterns, respectively. Next, we propose a self-paced intervention mechanism based on training status to create adversarial samples by reassembling variant patterns across neighborhoods and time stamps to remove the spurious impacts of variant patterns. Finally, we propose a sample importance reweighting strategy to distinguish invariant and variant patterns better via focusing on the key training samples. Extensive experiments on both synthetic and real-world dynamic graph datasets demonstrate the superiority of our proposed method over state-of-the-art baselines under distribution shifts.

## 1 INTRODUCTION

Dynamic graphs are ubiquitous in the real world, where some nodes and edges evolve along with time. Thanks to the power to capture both structural and temporal patterns simultaneously, dynamic graph neural networks (DGNNs) have achieved impressive successes in various real-world applications, including financial networks (Nascimento et al., 2021), social networks (Rossi et al., 2020b), traffic networks (Guo et al., 2021), etc. However, existing DGNNs have certain limitations when it comes to addressing distribution shift, which is common in practical scenarios (Holt, 2004; Zhang & Qi, 2005; Jin et al., 2021; Gao et al., 2021; Russell et al., 2019; Berk, 1983).

Existing research on out-of-distribution DGNNs primarily focuses on invariant learning, assuming the existence of invariant patterns across distributions (Zhang et al., 2022; Wu et al.). These studies incorporate the use of an additional loss component called invariant risk minimization (IRM) to capture invariances (Arjovsky et al., 2019). However, the fixed proportion of ERM and IRM in the training loss poses challenges for dynamically adjusting the training strategy in existing works. Moreover, the importance of diversity among samples has been overlooked when attempting to disentangle the invariant patterns, i.e., emphasizing samples that exhibit spurious correlations makes it difficult to capture invariance effectively. Therefore, adjusting the weight of different samples based on their properties and the training stage is crucial for capturing the invariant pattern successfully.

Curriculum learning (Bengio et al., 2009; Wang et al., 2021b) is a powerful approach that enables the dynamic adjustment of the training strategy, such as data reweighting and objection adjustment based on different training statuses. In this paper, we explore the application of curriculum learning

to address distribution shifts in dynamic graphs, which is a novel and unexplored area, and it poses the following challenges: (i) devising a customized evaluation strategy for training statuses, and (ii) designing a tailored strategy for reweighting the importance of samples, specifically to handle distribution shifts in dynamic graphs, e.g., the appearance or disappearance of graph nodes and edges.

To address these challenges, this paper presents a Curriculum Dynamic Graph Invariant Learning (**CDGIL**) model. The CDGIL model is introduced as a solution for handling distribution shifts in dynamic graphs by effectively capturing and utilizing both invariant and variant patterns. In particular, we first propose a dual disentangled dynamic attention network to capture the invariant and variant patterns, respectively. Furthermore, we propose a novel self-adaptive intervention method based on training status to create adversarial samples by reassembling variant patterns across neighborhoods and time stamps to remove the spurious impacts of variant patterns. Finally, we propose a novel curriculum-based sample importance reweighting method, which evaluates the importance of various data samples, and dynamically adjusts their weight in the training process. In summary, we made the following contributions:

- We investigate curriculum dynamic graph invariant learning for the first time, and propose a exquisitely designed curriculum learning method for dynamic graph generalization, which can automatically adjust both the training procedure and data weight according to data sample importance and different training stages.

- We design a self-adaptive intervention method to combine the invariant and variant parts, aiming to strengthen the generalization ability of our model and enhance our performance while facing data from unknown environments.

- We propose a novel curriculum-based sample importance reweighting method, by increasing the weight of data samples that are important to the current model and decreasing the weight of data samples that are not important to the current model, our models can get more suitable data inputs and get better performance optimization.

- Extensive experiments show that our proposed CDGIL method has the ability to outperform all of the state-of-the-art baselines on all of the dynamic graph datasets with distribution shifts.

## 2 RELATED WORKS

### 2.1 DYNAMIC GRAPH NEURAL NETWORKS

Dynamic Graph Neural Networks (DGNN) (Skarding et al., 2021; Zhu et al., 2022) have attracted intensive research attention as a powerful approach to handling the complex structural and temporal information in dynamic graphs. It has been applied to a wide range of real-world sceneries, such as action recognition (Yan et al., 2018), epidemic forecasting (Panagopoulos et al., 2020; Rozemberczki et al., 2021), social networks (Rossi et al., 2020b; Cai et al., 2022; Goyal et al., 2020), recommendation (Song et al., 2019), traffic prediction (Guo et al., 2021; Diao et al., 2019), and anomaly detection (Liu et al., 2021; Weber et al., 2019; Wang et al., 2021a). There are two mainstream DGNN methods, whose main difference between them is the order of dealing with time series and structural information. One of them utilizes a graph neural network (GNN) to aggregate structural information for the graph at each time at first and then adopts sequence models such as recurrent neural networks (RNN) (Yang et al., 2021; Sun et al., 2021; Hajiramezanali et al., 2019; Seo et al., 2018b; Pareja et al., 2020) or self-attention modules (Sankar et al., 2020) to process the temporal information. While the other method proposes to transform temporal links into a time-dependent function. using time-encoding techniques, then use GNN (Wang et al., 2021c; Cong et al., 2021; Xu et al., 2020; Rossi et al., 2020a) to process the graph with the time series information and get structural information. DIDA (Zhang et al., 2022) is the only work for dynamic graph distribution shift, it proposes a disentangled spatio-temporal attention network and a spatio-temporal intervention mechanism in order to handle spatio-temporal distribution shifts in dynamic graphs. But its training strategy is not related to the training status or sample importance.

## 2.2 Out-of-Distribution Generalization for Graph

Traditional machine learning method has the assumption that the training and test sets are guaranteed to be independently and identically distributed (i.i.d.), but this assumption does not hold in many real-world scenarios (Shen et al., 2021). Ignoring the fact that this assumption does not always hold may lead to the degradation of model performance (Fang et al., 2020). Various work has been done on this important question, such as IRM (Arjovsky et al., 2019), DRO (Rahimian & Mehrotra, 2019), REx (Krueger et al., 2021), and so on. In particular, designing models able to generalize in out-of-distribution (OOD) scenarios has attracted remarkable interest in graph representation learning. GIL (Li et al., 2022b) is proposed to capture the invariant relationships between predictive graph structural information and labels in a mixture of latent environments. OOD-GNN (Li et al., 2022a) proposes a nonlinear graph representation decorrelation method and a scalable global-local weight estimator to learn out-of-distribution (OOD) generalized graph representation under complex distribution shifts. However, most of the previous generalization work is only conducted on the static graph and failed to capture the feature of dynamic graphs such as timestamps.

## 2.3 Curriculum Learning

Curriculum learning (CL) (Bengio et al., 2009; Wang et al., 2021b; Soviany et al., 2022) is a popular method to train machine learning models in a meaningful order, such as from easier data to harder data, which is able to improve the performance of machine learning models, as well as bring faster convergence. Thanks to its powerful ability, curriculum learning have been widely applied to numerous branches of machine learning, including computer vision (Huang et al., 2020b), natural language processing (Cirik et al., 2016), speech (Braun et al., 2017), medical (Lotter et al., 2017), robotics (Florensa et al., 2017) and etc. Curriculum learning mainly includes two parts, difficulty measurer and training scheduler. The difficulty measurer aims to evaluate the difficulty of different input data, and the training scheduler can adjust the sequence according to the difficulty. Typically, there are two kinds of curriculum learning. One is predefined curriculum learning, or vanilla curriculum learning, where both the difficulty measurer and training scheduler is designed by human prior and expert domain knowledge. The other is automatic curriculum learning, where one or both difficulty measurer and training scheduler is learned from data-driven algorithms, which can change with different training procedures, rather than a prior decision. And there are many kinds of automatic curriculum learning, such as self-paced learning (Kumar et al., 2010), transfer learning (Weinshall et al., 2018), reinforcement learning (Florensa et al., 2017) and teacher-student method (Kim & Choi, 2018). However, the current curriculum learning methods failed to capture the dynamic feature, and thus can't be applied to dynamic graph out-of-distribution generalization.

# 3 Preliminary

In this section, we introduce the notations of dynamic graphs and their spatio-temporal distribution shift.

## 3.1 Dynamic Graph

Typically, a graph consists of nodes and edges: $\mathbf{G} = (\mathbf{V}, \mathbf{E})$, where $\mathbf{V}$ is the set of vertices and $\mathbf{E}$ is the set of edges. And in a dynamic graph, some graph nodes or graph edges will appear or disappear with the passing of time. So a dynamic graph can be formalized into the following form: $\mathbf{G} = \{\mathbf{G}_t : t = 1, 2, \cdots, T\}$, where $\mathbf{G}_t = (\mathbf{V}_t, \mathbf{E}_t)$, $t$ is the different time stamp, and $T$ is the total number of time stamps. $\mathbf{V} = \bigcup_{t=1}^{T} \mathbf{V}_t$ is the node set of dynamic graph, and $\mathbf{E} = \bigcup_{t=1}^{T} \mathbf{E}_t$ is the edge set of dynamic graph. The prediction task of the dynamic graph is predicting future labels using history graphs: $p(\mathbf{Y}_{t+1}|\mathbf{G}_1, \mathbf{G}_2, \cdots, \mathbf{G}_t) = p(\mathbf{Y}_{t+1}|\mathbf{G}_{1:t})$. In this paper, we mainly study node-level tasks, so $\mathbf{Y}_{t+1}$ typically represents the node property or edge status. And the prediction task can be formulated as a minimization task:

$$\min_{\theta} E_{y_{t+1} \sim p_{\text{train}}(\mathbf{y}_{t+1})} \mathbf{L}(f_\theta(\mathbf{G}_{1:t}), y_{t+1}), \tag{1}$$

where $y_{t+1}$ represents the instance of label, and $\mathbf{y}_{t+1}$ represents the abstract of label.

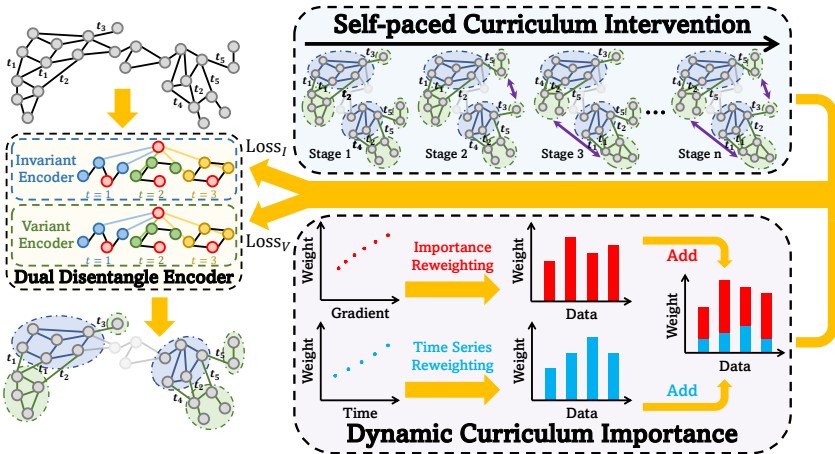

Figure 1: The framework of our proposed method **CDGIL**. Firstly, we adopt the dual disentangled dynamic attention network to capture the invariant patterns and variant patterns separately. Secondly, we conduct the self-paced intervention on dynamic graphs, in order to further enhance the generalization ability of the invariant encoder using variant patterns. Thirdly, we apply the dynamic curriculum method for importance reweighting, calculating the weights of data by evaluating its time as well as importance. Finally, we calculate the training loss, updating the invariant encoder and variant encoder, separately.

## 3.2 DISTRIBUTION SHIFT

However, learning a model for training distribution may suffer from a distribution shift between training distribution and testing distribution, which is still a key challenge. In our paper, we adopt the assumption that the relationship between dynamic graph data and the label remains the same in training distribution and testing distribution: $p_{train}(\mathbf{Y}_{t+1}|\mathbf{G}_{1:t}) = p_{test}(\mathbf{Y}_{t+1}|\mathbf{G}_{1:t})$ following (Wang et al., 2021c; Qiu et al., 2020; Huang et al., 2020a; Zhou et al., 2018; Trivedi et al., 2019). However, the distribution of the dynamic graph is different between training distribution and testing distribution: $p_{train}(\mathbf{G}_{1:t}) \neq p_{test}(\mathbf{G}_{1:t})$, i.e. our model is suffering covariate shift problem.

## 4 METHOD

To tackle the problem of spatio-temporal distribution shift in dynamic graphs, we propose our novel curriculum dynamic graph invariant learning in three parts. Firstly, in Section 4.1, we introduce our base model: dual disentangled dynamic attention network. Secondly, in Section 4.2, we introduce our proposed self-paced curriculum intervention method. Finally, in Section 4.3, we introduce our dynamic curriculum method for importance reweighting method.

### 4.1 DUAL DISENTANGLED DYNAMIC GRAPH ATTENTION NETWORKS

In this subsection, we introduce our base model: dual disentangled dynamic graph attention networks ($D^3$GAT). In previous work (Zhang et al., 2022), disentangled dynamic graph attention networks (DDGAT) compute the invariant patterns and variant patterns at the same time, limiting the ability of the variant patterns to adjust with the invariant patterns. Here we introduce it first.

Disentangled dynamic graph attention networks (DDGAT) apply the self-attention mechanism to integrate the node information of its neighborhood. The neighborhood of node $v$ at timestamp $t$ can be defined as: $N_t(v) = \{u : (v, u) \in \mathbf{E}_t\}$. And the history neighborhood of node $v$ at timestamp $t$ is defined as: $\widetilde{N}_t(v) = \bigcup_{\tau=1}^{t} N_\tau(v)$. To fully explore the neighborhood relationship and bring together all the information, the spatio-temporal graph attention mechanism aggregates all of the history neighbors of node $v$. And to express the different edges of different times, DDGAT also adopts $\mathbf{TE}(t)$ as time encoding of timestamp $t$. Thus the self-attention mechanism is carried out between the concatenation of node embedding as well as time encoding of node $v$ and all of its historic neighbors. Thus, we have query, key, and value as below, where $\mathbf{z}_v^t$ for the embedding of

node $v$ at time $t$:

$$\mathbf{q}_v^t = \mathbf{W}_q(\text{concat}(\mathbf{z}_v^t, \mathbf{TE}(t))), \mathbf{k}_u^{t'} = \mathbf{W}_k(\text{concat}(\mathbf{z}_u^{t'}, \mathbf{TE}(t'))), \mathbf{v}_u^{t'} = \mathbf{W}_v(\text{concat}(\mathbf{z}_u^{t'}, \mathbf{TE}(t'))).$$
$$(2)$$

And the query, key, and value are used to furthermore calculate the invariant and variant structural pattern masks. To further capture invariant patterns, a learnable mask $\mathbf{m}_f = \text{Softmax}(\mathbf{w}_f)$ is proposed to mask the variant feature. Then is the result of invariant and variant patterns:

$$\mathbf{z}_I^t = \text{Softmax}(\frac{\mathbf{q} \cdot \mathbf{k}^T}{\sqrt{d}})\mathbf{v} \cdot \mathbf{m}_f, \tag{3}$$

$$\mathbf{z}_V^t = \text{Softmax}(-\frac{\mathbf{q} \cdot \mathbf{k}^T}{\sqrt{d}})\mathbf{v}. \tag{4}$$

And after calculating the invariant and variant patterns, they are added up as the node feature input of the next layer:

$$\mathbf{z}_v^t = \mathbf{z}_I^t(v) + \mathbf{z}_V^t(v), \tag{5}$$

where the output vector $\mathbf{z}_v^t$ is for the next layer: layer $l+1$, while the input vectors $\mathbf{z}_I^t(v)$ and $\mathbf{z}_V^t(v)$ are from the current layer: layer $l$. With the aggregation and disentanglement of neighborhood information and invariant and variant pattern, the proposed disentangled dynamic graph attention networks with $L$ layers can represent the invariant patterns as well as the variant patterns in the $L$-th dynamic history neighborhood.

To summarize, the disentangled dynamic graph attention networks (DDGAT) get the input of the whole dynamic graph, and the output of DDGAT is the invariant pattern of the node feature as well as the variant pattern of the node feature. But as mentioned above, it jointly calculates the invariant pattern and the variant pattern of the node feature, which limits pattern learning, because they are decided jointly, and can not adjust with regard to the other.

In our method, we adopt dual DDGAT, utilizing two modules for recognizing the invariant pattern and the variant pattern, separately. We call the two module invariant encoder and variant encoder for short. Invariant encoder and variant encoder have different learning objectives. The invariant encoder aims to precisely discover the invariant pattern of graphs, while the variant encoder captures the variant pattern of graphs in order to improve the generalization ability of the invariant encoder. The pattern captured by the invariant encoder and variant encoder is presented as $\mathbf{z}_I$ and $\mathbf{z}_V$, respectively.

## 4.2 SELF-PACED CURRICULUM INTERVENTION

The objective of the invariant encoder is to capture the invariant patterns precisely, and the variant patterns are a good source to let the invariant encoder generalize better. The adversarial variant patterns are reassembled with the invariant pattern to conduct intervention and the intensity of intervention is decided by the training stage.

The reassembled result is presented as $\mathbf{z}_r$, and the result is controlled by a parameter reassemble $\lambda_r$, where the larger the $\lambda_r$ is, the greater the intensity of the intervention result is:

$$\mathbf{z}_r = \text{Assemble}(\mathbf{z}_I, \mathbf{z}_V, \lambda_r), \tag{6}$$

where the assemble function is calculated from the intervention result of the variant part and invariant part. Following (Zhang et al., 2022), we adopt the approximate intervention method, randomly replacing a specific variant pattern with another random variant pattern:

$$\mathbf{z}_I^{t_1}(u), \mathbf{z}_V^{t_1}(u) \leftarrow \mathbf{z}_I^{t_1}(u), \mathbf{z}_V^{t_2}(v). \tag{7}$$

The $\lambda_r$ is the parameter controlling the intervention intensity. In the early training stage, the invariant and variant encoder are not well-trained, thus the $\lambda_r$ should be relatively smaller. While in the late training stage, the invariant and variant encoder are well-trained in training distribution and eager to generalize to testing distribution or other unknown distributions. So this time, the $\lambda_r$ should be relatively larger. In our proposed self-paced curriculum intervention method, we adjust the $\lambda_r$ with our training stage and evaluate the training stage through the functional loss $\ell$. We set a threshold for the minimum loss $\ell_{min}$ for each different task, and calculate the $\lambda_r$ for each epoch:

$$\lambda_r = \lambda_0 * \min(\ell, \ell_{min})^{-1}. \tag{8}$$

To summarize, for the invariant encoder's loss, we add the training loss $\ell_{trainI}$ to the loss of reassembled with adversarial variant patterns, $\ell_{assemble}$. Thus, we have our invariant loss for our invariant encoder.

$$\ell_I = \ell_{trainI} + \ell_{assemble} * \lambda_{assemble}, \tag{9}$$

where $\lambda_{assemble}$ is the weight assigned to $\ell_{assemble}$, because it's not so important as the loss of training.

The objective of the variant encoder is to capture the variant patterns in order to enhance the generalization ability of the invariant encoder. And we maximize the difference between variant patterns reassembled with the invariant patterns and the invariant patterns themselves, further enhancing the variation between variant patterns and invariant patterns, formatting adversarial variant patterns.

To capture adversarial variant patterns compared to invariant patterns, we evaluate the difference between the reassembled patterns and the invariant patterns. In this paper, we utilize the mean of the square of the difference between them ($\mathbf{z}_r$ and $\mathbf{z}_I$):

$$\text{Diff}_{var} = \text{mean}((\mathbf{z}_r - \mathbf{z}_I)^2). \tag{10}$$

On the basis of the training loss of the variant decoder, we subtract our calculated difference variant patterns made from the training loss. So the final loss of the variant encoder is presented below:

$$\ell_V = \ell_{trainV} - \text{Diff}_{var} * \lambda_{diff}, \tag{11}$$

where $\lambda_{diff}$ is the weight assigned to $\text{Diff}_{var}$.

In our method, the capture of invariant patterns should adjust with different training stages, and the capture of variant patterns should adjust with different invariant patterns. So the optimization of our proposed dual disentangled dynamic attention network is of two steps. In the first step, we calculate the loss of the invariant encoder and optimize its weight while freezing the weight of the variant encoder. And in the second step, we calculate the loss of the variant encoder and optimize its weight while freezing the weight of the invariant encoder. Our two-step optimization can capture invariant patterns more precisely and capture proper variant patterns.

### 4.3 Dynamic Curriculum Importance

#### 4.3.1 Curriculum Time Serial Reweighting

In dynamic graphs, all the nodes and edges are related to a certain timestamp. And the data of different timestamps tend to have different features. With the smaller gap in time, the graph data tend to have more similar properties. To further consider the time information of the graph, we propose the curriculum time serial reweighting method for dynamic graphs.

More specifically, we change the weight of the data sample of the graph with timestamp $t$: $\mathbf{w}_t = (1 + \lambda)^t$. And for all of the data, we adjust their weight to maintain that the average weight of all the data remains 1, in order to avoid further introducing bias:

$$\mathbf{w}_{time} = \mathbf{w}_t + (1 - \text{average}(\mathbf{W}_t)), \tag{12}$$

where $\mathbf{W}_t$ is the vector of all the dynamic graph data according to weights, consisting of each $w_t$ of data samples.

#### 4.3.2 Curriculum Sample Importance Reweighting

In curriculum learning, one of the basic assumptions is that different data should not be treated equally, i.e. different data could have different weights or order. However, the traditional method for dynamic graph out-of-distribution generalization treats all of the data in the same way. Thus here we discuss how to distinguish the importance of different data, and how to change their weights in order to gain better results and generalization ability.

During experiments, we discover that the gradient is an important signal of the data's importance to the model now. We made the trial that observed the pattern of the gradient of the prediction results, and we discover that with the entanglement of the variant features and invariant features, the gradient descent failed to optimize the model to fit the training set or enhance the generalization ability. So we propose our novel curriculum learning for sample reweighting through the gradient measure.

For data samples $D = \{d_1, d_2, \cdots, d_n\}$, we first calculate their loss $\ell(\mathbf{y}_{pred}, \mathbf{y}_{true})$, and perform gradient backward, thus we have the gradient of all of the prediction results: $\text{grad}(d_1), \text{grad}(d_2), \cdots,$ $\text{grad}(d_n)$. Using the gradient backward result of the prediction results, we can reweight our data's weight via a function $f$ mapping gradients to weights:

$$\mathbf{w}_{imp}^i = f(\text{grad}(d_i)). \tag{13}$$

And the mapping function is a two-stage segmentation function. For the gradient that is smaller than zero, the function is $f(x) = 1$, which will increase the weight of their data samples. And for the gradient that is larger than zero, the function is $f(x) = -\exp(\text{sigmoid}(\log(x) * A + B) * C + D)$, where the $A$, $B$, $C$ and $D$ are hyper-parameters. This mapping function will increase the weight of the data samples that have a negative gradient, slightly decrease the data samples that have a large positive gradient, and largely decrease the data samples that have small positive samples. This mapping function will reweight their weight by evaluating the importance and further have better optimization and generalization ability.

After calculating the weights of the time series as well as sample importance, we add them together for the final output for each data sample $\mathbf{w}_{time} + \mathbf{w}_{imp}$. However, there appear some cases while applying these weights that the weights drop below zero and resulting in loss $\ell_{train}$ drop below zero. Thus we adopt a soft positive function on the weights in order to avoid optimization of the contradictory direction:

$$\mathbf{w} = \max(\mathbf{w}_{time} + \mathbf{w}_{imp}, 0). \tag{14}$$

The final reweighting weight $w$ is used to calculate the training loss $\ell_{train}$. And the overall algorithm of our proposed CDGIL method is shown in Table1.

---

**Algorithm 1** Our proposed curriculum dynamic graph invariant learning method

---

1: **Initialize.** Number of training epochs $E$, dynamic graph dataset $G$, invariant decoder $D_I$, variant decoder $D_V$.
2: **for** $e = 1 : E$ **do**
3:     Calculate $z_I$ and $z_V$: $z_I^t = \text{Softmax}(\frac{q \cdot k^T}{\sqrt{d}})v \cdot m_f, z_V^t = \text{Softmax}(-\frac{q \cdot k^T}{\sqrt{d}})v$.
4:     Calculate $w_{time}$ and $w_{imp}$: $w_{time} = w_t + (1 - \text{average}(W_t)), w_{imp}^i = f(\text{grad}(d_i))$.
5:     Calculate $w$: $w = \max(w_{time} + w_{imp}, 0)$.
6:     Calculate invariant loss $\ell_I = \ell_{trainI} + \ell_{assemble} * \lambda_{assemble}$.
7:     Perform gradient descent on the invariant encoder.
8:     Calculate variant loss $\ell_V = \ell_{trainV} - \text{Diff}_{var} * \lambda_{diff}$.
9:     Perform gradient descent on the variant encoder.
10: **end for**
11: **Output.** The well-trained model.

---

## 5 EXPERIMENTS

In this section, we present the experiments to verify the effectiveness and wide applicability of our proposed method, including experimental setup, results, and ablation studies. Please refer to Appendix for more details.

### 5.1 EXPERIMENTAL SETUP

**Baselines.** In the experiments, we consider three kinds of representative baselines, including (1) static GNNs: **GAE** (Kipf & Welling, 2016b) and **VGAE** (Kipf & Welling, 2016b), (2) dynamic GNNs: **GCRN** (Seo et al., 2018a), **EGCN** (Pareja et al., 2020), **DySAT** (Sankar et al., 2020), and (3) OOD generalization methods: **IRM** (Arjovsky et al., 2019), **GroupDRO** (Sagawa et al., 2019), **V-REx** (Krueger et al., 2021), **DIDA** (Zhang et al., 2022).

**Datasets.** The experiments are conducted on both real-world and synthetic dynamic graph datasets.

- **COLLAB** (Tang et al., 2012) is a dataset of cross-domain collaboration recommendations, including the authors and publications from 1990 to 2005. The nodes of the dynamic graph are authors, and the edges between nodes are coauthorship. Its edges contain five

Table 1: The experiment results (ROCAUC%) of different methods on real-world link prediction datasets. The best results are in bold. 'w/o DS' and 'w/ DS' denote test data with and without distribution shift.

| Model | COLLAB | | Yelp | |
|---|---|---|---|---|
| Test Data | w/o DS | w/ DS | w/o DS | w/ DS |
| GAE | $77.15_{\pm0.50}$ | $74.04_{\pm0.75}$ | $70.67_{\pm1.11}$ | $64.45_{\pm5.02}$ |
| VGAE | $86.47_{\pm0.04}$ | $74.95_{\pm1.25}$ | $76.54_{\pm0.50}$ | $65.33_{\pm1.43}$ |
| GCRN | $82.78_{\pm0.54}$ | $69.72_{\pm0.45}$ | $68.59_{\pm1.05}$ | $54.68_{\pm7.59}$ |
| EGCN | $86.62_{\pm0.95}$ | $76.15_{\pm0.91}$ | $78.21_{\pm0.03}$ | $53.82_{\pm2.06}$ |
| DySAT | $88.77_{\pm0.23}$ | $76.59_{\pm0.20}$ | $78.87_{\pm0.57}$ | $66.09_{\pm1.42}$ |
| IRM | $87.96_{\pm0.90}$ | $75.42_{\pm0.87}$ | $66.49_{\pm10.78}$ | $56.02_{\pm16.08}$ |
| VREx | $88.31_{\pm0.32}$ | $76.24_{\pm0.77}$ | $\underline{79.04_{\pm0.16}}$ | $66.41_{\pm1.87}$ |
| GroupDRO | $88.76_{\pm0.12}$ | $76.33_{\pm0.29}$ | $\mathbf{79.38_{\pm0.42}}$ | $66.97_{\pm0.61}$ |
| DIDA | $\underline{91.97_{\pm0.05}}$ | $\underline{81.87_{\pm0.40}}$ | $78.22_{\pm0.40}$ | $\underline{75.92_{\pm0.90}}$ |
| Ours | $\mathbf{93.60_{\pm0.11}}$ | $\mathbf{84.39_{\pm0.54}}$ | $77.15_{\pm1.54}$ | $\mathbf{76.44_{\pm1.79}}$ |

Table 2: The experiment results (ROCAUC%) of different methods on synthetic link prediction datasets. The best results are in bold. 'w/o DS' and 'w/ DS' denote test data with and without distribution shift.

| Model | synthetic-0.4 | | synthetic-0.6 | | synthetic-0.8 | |
|---|---|---|---|---|---|---|
| Split | Train | Test | Train | Test | Train | Test |
| GCRN | $69.60_{\pm1.14}$ | $72.57_{\pm0.72}$ | $74.71_{\pm0.17}$ | $72.29_{\pm0.47}$ | $75.69_{\pm0.07}$ | $67.26_{\pm0.22}$ |
| EGCN | $78.82_{\pm1.40}$ | $69.00_{\pm0.53}$ | $79.47_{\pm1.68}$ | $62.70_{\pm1.14}$ | $81.07_{\pm4.10}$ | $60.13_{\pm0.89}$ |
| DySAT | $84.71_{\pm0.80}$ | $70.24_{\pm1.26}$ | $89.77_{\pm0.32}$ | $64.01_{\pm0.19}$ | $94.02_{\pm1.29}$ | $62.19_{\pm0.39}$ |
| IRM | $85.20_{\pm0.07}$ | $69.40_{\pm0.09}$ | $89.48_{\pm0.22}$ | $63.97_{\pm0.37}$ | $\mathbf{95.02_{\pm0.09}}$ | $62.66_{\pm0.33}$ |
| VREx | $84.77_{\pm0.84}$ | $70.44_{\pm1.08}$ | $89.81_{\pm0.21}$ | $63.99_{\pm0.21}$ | $94.06_{\pm1.30}$ | $62.21_{\pm0.40}$ |
| GroupDRO | $84.78_{\pm0.85}$ | $70.30_{\pm1.23}$ | $89.90_{\pm0.11}$ | $64.05_{\pm0.21}$ | $94.08_{\pm1.33}$ | $62.13_{\pm0.35}$ |
| DIDA | $\underline{87.92_{\pm0.92}}$ | $\underline{85.20_{\pm0.84}}$ | $\underline{91.22_{\pm0.59}}$ | $\underline{82.89_{\pm0.23}}$ | $92.72_{\pm2.16}$ | $\underline{72.59_{\pm3.31}}$ |
| Ours | $\mathbf{89.77_{\pm0.15}}$ | $\mathbf{87.60_{\pm0.18}}$ | $\mathbf{91.88_{\pm0.26}}$ | $\mathbf{84.67_{\pm0.27}}$ | $\underline{94.84_{\pm0.29}}$ | $\mathbf{79.54_{\pm0.87}}$ |

sub-domains: data mining, medical informatics, theory, visualization, and database. Here we choose data mining as the testing domain, and the other four sub-domains as the training domains.

- **Yelp** (Sankar et al., 2020) is a dataset of businesses and reviews, including the customers and their reviews on businesses. The nodes of the dynamic graph are customers or businesses, and the edges between them are reviews. Its edges contain five sub-domains: pizza, American food, coffee & tea, sushi bars, and fast food. Here we choose pizza as the testing domain, and the other four sub-domains as the training domains.

- **Synthetic Dataset** (Zhang et al., 2022) is one manually designed dataset based on collab dataset, introducing external node features to create distribution shifts. The external node features are related to the average positive sample rate $\bar{p}$, wherein the testing dataset, $\bar{p}$ is set to 0.1, and in the training dataset, $\bar{p}$ is set to 0.4, 0.6 or 0.8, formatting the three datasets: synthetic-0.4, synthetic-0.6, and synthetic-0.8, where we can see that the larger $\bar{p}$ is, the larger the distribution shift between the training part and the testing part of the dataset.

## 5.2 RESULTS

- The performances of the baseline methods drop significantly under distribution shifts, although some of them show relatively competitive results on test data without distribution shifts. For example, when existing distribution shifts the performance of DySAT, which is one representative dynamic method, drops more than 10% in COLLAB and Yelp datasets. It shows that the existing GyGNNs fail to handle distribution shifts and focus on variant patterns to make predictions, leading to poor OOD generalization. As one dynamic graph OOD generalization method, DIDA shows strong performances to handle distribution shifts

but the performance drop is also significant. It means that although this method attempts to capture invariant patterns and only uses them to make predictions, the result is not promising since it ignores the training status and sample importance when learning invariant patterns in dynamic graphs.

- Our method can accurately capture invariant patterns in dynamic graphs to consistently remove the impact of variant patterns under distribution shifts. For the real-world datasets, we can find consistent performance improvements. And for the synthetic datasets, we can also observe that our method achieves the most stable performances as the shift level increases, while almost all baselines increase in train results and decline in test results. It verifies that the existing baselines can easily exploit variant patterns for predictions and suffer from their harmful effects on OOD generalization.

### 5.3 ABLATION STUDIES

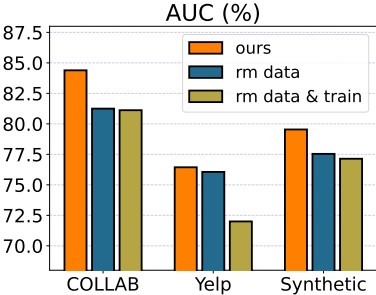

Figure 2: The ablation study result of our method compared to our method without dynamic curriculum importance sample reweighting, and our method without dynamic curriculum importance sample reweighting as well as self-paced curriculum intervention.

We first remove the dynamic curriculum importance sample reweighting module. And we can observe that the performance gained a sharp drop without our novel sample reweighting method, demonstrating its effectiveness.

Next, we also consider removing the self-paced curriculum intervention as well as the dynamic curriculum importance sample reweighting. And we can observe that the performance drop without our novel training scheduler method, which indicates the effectiveness of our designs.

### 5.4 COMPLEXITY ANALYSIS

Here we will analyze our proposed CDGIL method's computational complexity. Let $|V|$ and $|E|$ denote the number of total nodes and edges, respectively. Let $d$ denote the dimension of the hidden representation. From (Zhang et al., 2022), we learn that the computational complexity of the disentangled DGNN is $O(|E| d + |V| d^2)$. Further, let $|E_t|$ denote the number of edges to train and test, for each training step, the computation cost of curriculum learning for data is only several times of $|E_t|$, or $O(|E_t|)$. And the computation cost of self-adaptive curriculum learning for intervention is $|E| d$, where we conduct intervention operation on the whole graph. What's more, we can also find that $O(|E_t|)$ is dominated by $O(|E|)$. Finally, we get our result: the computational complexity of our proposed CDGIL is $O(|E| d + |V| d^2)$, less than DIDA's, which is $O(|E| d + |V| d^2 + |E_t| |S| d)$, and our method outperforms DIDA largely in all of the datasets.

### 6 CONCLUSION

In this paper, we propose Curriculum Dynamic Graph Invariant Learning (CDGIL) to handle the dynamic graph distribution problem. More specifically, we present to capture invariant and variant patterns directed by our proposed curriculum learning method considering both training status and sample importance. Firstly, we utilize dual disentangled dynamic attention networks to capture invariant and variant patterns and optimize them separately. Then, we conduct self-paced curriculum intervention to generalize better. Finally, we propose to compute the weight of data samples by evaluating their time series weight as well as sample importance weight. Extensive experiments and ablation studies further demonstrate the effectiveness of our proposed method.

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

# A  NOTATIONS

| Notations | Descriptions |
|:---:|:---|
| $\mathbf{G} = (\mathbf{V}, \mathbf{E})$ | A graph consists of its node set as well as its edge set |
| $\mathbf{G}_t = (\mathbf{V}_t, \mathbf{E}_t)$ | Graph at time $t$ |
| $\mathbf{Y}_{t+1}$ | The feature labels at time $t+1$ |
| $\mathbf{G}_{1:t}$ | The set of graph from time 1 to time $t$ |
| $p_{\text{train}}$ | The distribution in training set |
| $p_{\text{test}}$ | The distribution in testing set |
| $y_{t+1}$ | A feature label at time $t+1$ |
| $\mathbf{y}_{t+1}$ | Feature labels at time $t+1$ |
| $f_\theta(\cdot)$ | The prediction function |
| $N_t(v)$ | The neighborhood of node $v$ at time $t$ |
| $\widetilde{N}_t(v)$ | The neighborhood of node $v$ from time 1 to time $t$ |
| $TE(\cdot)$ | The time encoding |
| $\mathbf{z}_v^t$ | The embedding of node $v$ at time $t$ |
| $\mathbf{q}, \mathbf{k}, \mathbf{v}$ | The query, key and value vector |
| $\mathbf{z}_I^t(v), \mathbf{z}_V^t(v)$ | The invariant and variant patterns of node $v$ at time $t$ |
| $\mathbf{m}_f$ | The learnable mask for invariant pattern |
| $\ell_I, \ell_V$ | Invariant loss and variant loss |
| $\mathbf{w}_{time}, \mathbf{w}_{imp}$ | Weight of time and weight of importance |

# B  REPRODUCIBILITY DETAILS

## B.1  DATASET DETAILS

We provide the statistic of our datasets in Table 3.

Table 3: Statistics of dynamic graph datasets.

| Dataset | #Timestamps | #Nodes | #Edges | Time Interval | Node Feature Dimension |
|:---|:---:|:---:|:---:|:---:|:---:|
| COLLAB | 16 | 23035 | 151790 | year | 32 |
| Yelp | 24 | 13095 | 65375 | month | 32 |
| Synthetic | 16 | 23035 | 151790 | year | 64 |

For Collab and Yelp datasets, we adopt word2vec? to extract a feature vector with a length of 32 from each paper abstract or business review and use the mean pooling to get the feature of an author or a user as the node feature of dynamic graphs. And for the Synthetic dataset, we add the node feature computed in Section 5.1 to concat with the original node feature while inheriting the topology of the original dynamic graph, thus the length of the node feature is 64. And for Collab and Synthetic datasets, we split the 16 timestamps into 3 groups: 10, 1, and 5 for training, validation, and testing separately. For the Yelp dataset, we split the 24 timestamps into 3 groups: 20, 2, and 8 for training, validation, and testing separately.

## B.2  CDGIL DETAILS

**Basic Details.** We set the dimension of the hidden layer to 16, and the number of layers to 2. As for the optimizer, we adopt Adam? with a learning rate 0.01 and weight decay $5 \times 10^{-7}$. The number of total epochs is set to 1000, and we also adopt an early stop method. And for evaluation, we adopt random negative sampling to get negative edges between nodes without edges, where we set the same random seed to guarantee fairness. We predict the link by multiplying the two learned nodes embedding together. And the evaluation metric is the area under the roc curve (AUCROC).

**Curriculum Details.** For self-paced curriculum intervention, the intensity parameter $\lambda_r$ is decided by the $\lambda_0$ as well as the loss $\ell$. For all datasets, $\ell_{min}$ is set to 0.1, $\lambda$, and $\lambda_0$ is set to 0.015 for

COLLAB and 0.005 for others. And for the variant decoder, $\lambda_{diff}$ is also set to 0.05. For time serial reweighting, $\lambda$ is set to 0.001. While for sample importance reweighting, we set the parameter $A = 0.8$, $B = 0$, $C = -0.0004$, and $D = -9 + \lambda_D * epoch - \log(\lambda_{imp})$, where epoch refers to the corresponding epoch, $\lambda_D = 0.004$, and $\lambda_{imp}$ is the parameter multiplied to $\mathbf{w}_{imp}$, which is set to 0.15 for COLLAB dataset, and 0.05 for Yelp and Synthetic datasets.

### B.3 CONFIGURATIONS

All of our experiments are conducted in the following environments:

- **Operating System.** Ubuntu 20.04.5 LTS
- **CPU.** Intel(R) Xeon(R) Gold 6240 CPU @ 2.60GHz
- **GPU.** NVIDIA GeForce RTX 3090
- **Software.** Python 3.9.13, Cuda 11.7, PyTorch 1.11.0, PyTorch Geometric 2.0.4

## C   ADDITIONAL EXPERIMENTS

### C.1   HYPERPARAMETER SENSITIVITY

We carry out the hyperparameter sensitivity experiments. We can see that, either the hyperparameter is too big or too small will bring a performance drop, which indicates that too small will lose its effect, while too big will break the normal data weight.

| Dataset | 0.03 | 0.06 | 0.3 | 0.6 | Best baseline |
|---|---|---|---|---|---|
| COLLAB | 81.49 ± 0.77 | 83.83 ± 0.40 | 84.25 ± 0.30 | 83.69 ± 0.44 | 81.87 ± 0.40 |

Table 4: Hyperparameter sensitivity experiments on $\lambda_{imp}$.

| Dataset | 0.01 | 0.02 | 0.1 | 0.2 | Best baseline |
|---|---|---|---|---|---|
| Yelp | 76.84 ± 1.05 | 75.13 ± 1.95 | 76.41 ± 1.23 | 76.50 ± 2.09 | 75.92 ± 0.90 |
| Synthetic | 86.00 ± 0.32 | 85.64 ± 0.43 | 86.60 ± 0.48 | 86.98 ± 0.29 | 85.20 ± 0.84 |

Table 5: Hyperparameter sensitivity experiments on $\lambda_{imp}$.

| Dataset | 0.6 | 0.7 | 0.8 | 0.9 | Best baseline |
|---|---|---|---|---|---|
| COLLAB | 84.42 ± 0.26 | 84.60 ± 0.16 | 84.40 ± 0.43 | 83.24 ± 0.78 | 81.87 ± 0.40 |

Table 6: Hyperparameter sensitivity experiments on $A$.

### C.2   EXTENSIVE EXPERIMENTS

To validate the effectiveness of dual disentangled dynamic attention network, we compare its experimental result with disentangled dynamic attention network. Results are shown in Table 9.

To validate the effectiveness of CDGIL, we further add several representative baselines: GCN (Kipf & Welling, 2016a), GIN (Xu et al., 2018), GAT (Veličković et al., 2017), GraphSage (Hamilton et al., 2017), EERM (Wu et al., 2022), SR-GNN (Zhu et al., 2021), and results are shown in Table 10

| Dataset | -0.0003 | -0.0004 | -0.0005 | Best baseline |
|---|---|---|---|---|
| COLLAB | 84.54 ± 0.27 | 84.35 ± 0.30 | 84.53 ± 0.39 | 81.87 ± 0.40 |

Table 7: Hyperparameter sensitivity experiments on $C$.

| Dataset | 0.003 | 0.004 | 0.005 | Best baseline |
|---|---|---|---|---|
| COLLAB | 84.54 ± 0.27 | 84.35 ± 0.30 | 84.53 ± 0.39 | 81.87 ± 0.40 |

Table 8: Hyperparameter sensitivity experiments on $\lambda_D$.

| Dataset | D3GAT | DDGAT |
|---|---|---|
| COLLAB | 78.28 ± 0.40 | 77.25 ± 0.36 |
| Yelp | 66.51 ± 8.06 | 64.66 ± 5.28 |

Table 9: Comparison experiment between D3GAT and DDGAT.

| Dataset Test Data | COLLAB w/o DS | COLLAB w/ DS | Yelp w/o DS | Yelp w/ DS |
|---|---|---|---|---|
| GCN | 79.20 ± 0.08 | 69.24 ± 0.20 | 56.99 ± 0.18 | 64.18 ± 0.61 |
| GIN | 66.55 ± 0.32 | 60.65 ± 0.47 | 68.97 ± 0.95 | 67.11 ± 5.21 |
| GAT | 71.66 ± 0.02 | 65.79 ± 0.11 | 60.16 ± 0.78 | 65.14 ± 2.38 |
| Sage | 76.49 ± 0.30 | 66.19 ± 0.56 | 69.30 ± 0.18 | 68.67 ± 0.31 |
| EERM | OOM | OOM | 73.01 ± 3.99 | 69.92 ± 9.59 |
| SR-GNN | 79.36 ± 0.13 | 69.65 ± 0.47 | 70.41 ± 0.52 | 69.31 ± 0.30 |
| Ours | 93.60 ± 0.11 | 84.39 ± 0.54 | 77.15 ± 1.54 | 76.44 ± 1.79 |

Table 10: Comparison experiment between D3GAT and DDGAT.

