# OpenReview forum: "Curriculum Dynamic Graph Invariant Learning under Distribution Shift"
_ICLR.cc/2024/Conference — Submitted to ICLR 2024_

### Official Review · Reviewer_GMx5 · 2023-10-29

**Soundness:** 2 fair
**Presentation:** 2 fair
**Contribution:** 1 poor
**Rating:** 3
**Confidence:** 4

**Summary:**

This paper introduces a Curriculum Dynamic Graph Invariant Learning (CDGIL) model to handle the distribution shift problem in dynamic graphs. Specifically, the authors propose a dual disentangled dynamic attention network to capture both the invariant and variant patterns with a curriculum learning strategy. Experiments on both synthetic and real-world dynamic graphs are conducted to validate the proposed method.

**Strengths:**

1. The literature review on Dynamic Graph Neural Networks and OOD generalization seems to be thorough.

2. Experiments on both synthetic and real-world datasets are conducted.

**Weaknesses:**

1. D3GAT is proposed as an adaptation of DDGAT to overcome the issue that DDCAT “jointly calculates the invariant pattern and the variant pattern of the node feature, which limits pattern learning, because they are decided jointly and cannot adjust with regard to the other.” This claim seems to be unjustified. Through reading the paper, the reviewer is not convinced that calculating invariant pattern and the variant pattern would harm the model generalization ability. Some theoretical justifications or intuition demonstrations may be helpful.

2. The proposed strategy to calculate weights for samples from different time stamps seems to be trivial. Crucial details are missing in the methodology part. For example, it is unclear how W_t is decided in equation (12). Also, it is unclear how the weight, w, participated in the training process and calculation of loss in algorithm 1.

3. DDGAT should be a crucial baseline to be compared with, but it is completely excluded from the experimental evaluation part of the main paper. Only the performance of DDGAT on COLLAB and Yelp datasets is provided in the appendix part. It is left unexplained why DDGAT is not included for comparison on other datasets. Besides, the D3GAT only outperforms COLLAB by 1%, and its improvement on Yelp also does not seem to be significant given the large standard deviations.

4. The paper needs to be carefully proofread to fix all typos and meet the standards of academic writing. See minor issues for details. Note that the listed issues are not exhaustive.



Minor issues:

1. Section 2.3: It is incorrect to say “and etc”, “and” is redundant.

2. The title of table 2 is wrong.

3. The title of table 10 is wrong.

4. Table 1,2: there needs to be space between titles and tables.

5. Abbreviations like “can’t” should be avoided in formal writing.

**Questions:**

1. In section 3.2, it is mentioned that the models suffer from covariate shift problems. However, the datasets used in the experiment part contain training sets and test sets from different domains. Can they also experience a concept shift?

2. Regarding Table 1, the paper mentions that training sets and test sets are split into different subdomains. How are the experiments conducted for the “w/o DS” option where there is no distribution shift?

3. Are there any specific reasons to compare the results on training datasets in Table 2?

---

### Official Review · Reviewer_QAsp · 2023-11-01

**Soundness:** 3 good
**Presentation:** 3 good
**Contribution:** 3 good
**Rating:** 6
**Confidence:** 4

**Summary:**

This paper first applies curriculum learning to address distribution shifts in dynamic graphs and proposes a Curriculum Dynamic Graph Invariant Learning (CDGIL) model to handle distribution shifts in dynamic graphs by effectively capturing and utilizing both invariant and variant patterns. Firstly, the authors bring out a novel structure called dual disentangled dynamic graph attention networks based on the disentangled dynamic graph attention networks module(DDGAT) from DIDA[1]. The main difference between the novel structure and DDGAT is that the novel structure utilizes two modules for recognizing the invariant pattern and the variant pattern, separately. Then the authors propose a self-adaptive intervention method to combine the invariant and variant parts and improve the performance on unknown data. What's more, the authors also propose a novel curriculum-based sample importance reweighting method to adjust the weight of samples dynamically. The experiment and ablation studies demonstrate the effectiveness of the model in handling distribution shifts in dynamic graphs.

[1] Zeyang Zhang, Xin Wang, Ziwei Zhang, Haoyang Li, Zhou Qin, and Wenwu Zhu. Dynamic graph neural networks under spatio-temporal distribution shift. In Advances in Neural Information Processing Systems, 2022.

**Strengths:**

1. This paper addresses a common and important aspect of dynamic graph learning.
2. The background and motivation are presented nicely and clearly.
3. Comprehensive experimental results demonstrate the effectiveness of the model.
4. Organized and easy to follow.

**Weaknesses:**

1. Some symbols and nouns are missing from the labeling, for example:
   1. ERM mentioned in the Introduction
   2. t' in the key and value formula
   3. f in a learnable mask
   4. T in the result of invariant and variant patterns. Is the T for moment or exponent?
   5. d in the result of invariant and variant patterns
2. The framework diagram lacks the necessary symbols, for example, Z for each encoder and the output. Also, the loss of each part should also be clearly expressed in the framework diagram. The existing framework diagrams don't give a very clear picture of the framework.
3. More explanation is needed to explain the difference between DDGAT and dual DDGAT at the formula level. Is the final output vector any different? Or is the only difference between this dual DDGAT and DDGAT separate at the loss function level?
4. The loss functions are not specifically presented.
5. There should be a distinction between the d used in data samples in the CURRICULUM SAMPLE IMPORTANCE REWEIGHTING section and the d in the result of invariant and variant patterns
6. Lack of theoretical support for the design of the mapping function in the CURRICULUM SAMPLE IMPORTANCE REWEIGHTING section.
7. Unclear legend labeling for ABLATION STUDIES

**Questions:**

1. After reading the paper, I would like to ask about is the final output vector any different between dual DDGAT and DDGAT. Or is the only difference between this dual DDGAT and DDGAT at the loss function level?
2. DDGAT should be included in ABLATION STUDIES, not in the appendix because the compare between DDGAT and dual DDGAT is a very important part of the model.

---

### Official Review · Reviewer_GHD2 · 2023-11-03

**Soundness:** 3 good
**Presentation:** 2 fair
**Contribution:** 3 good
**Rating:** 3
**Confidence:** 4

**Summary:**

This paper studies the problem of distribution shifts in dynamic graphs. The authors propose CDGIL, which captures and utilizes invariant and variant patterns based on training status and sample importance, to tackle distribution shifts in dynamic graphs. The paper also experimentally evaluates the proposed solutions on both synthesized and real-world networks.

**Strengths:**

S1. This paper is the first to use curriculum learning to address the distribution shift problem in dynamic graphs.

S2. The idea of designing a dual disentangled dynamic attention network to capture two patterns is simple and intuitive.

**Weaknesses:**

W1. The paper is hard to follow due to several unclear presentations. For example, the implementation of the Assemble function in Equation 6 lacks clarity. The specific nature of $\ell_{trainI}$ and $\ell_{trainV}$ in Equation 9 and 11 is not elucidated. Additionally, it is not clear how the function f in Equation 13 transforms gradient vectors into scalar measures of importance.

W2. In Section 4.3.2, the authors indicate that the gradient is computed using the loss, which is then used to calculate the importance weights. These weights are subsequently used to reweight the loss. Given this process, there appears to be a potential inconsistency in Algorithm 1 where the weight calculation in steps 4-5 and the loss calculation in steps 6-9 may be out of sequence. If my understanding is correct, steps for gradient calculation and gradient zeroing should be inserted before steps 4-5.

W3. The importance sampling method described in Section 4.3 resembles a technique designed to accelerate convergence, and its relevance to curriculum learning does not appear to be particularly robust.

W4. It was found in Section 5.4 that DIDA is more complicated than this paper $|E_t||S|d$, which refers to the time complexity of the intervention algorithm. Based on my understanding, this paper uses an adaptive intervention coefficient, which adaptively adjusts the number of interventions and also has the corresponding time complexity. However, the paper does not provide specific details on the implementation of the Assemble function. My understanding may be incorrect. I hope the author can explain this in more detail.

**Questions:**

See W1-W4 for details

---

### Official Review · Reviewer_wfAZ · 2023-11-05

**Soundness:** 2 fair
**Presentation:** 2 fair
**Contribution:** 2 fair
**Rating:** 3
**Confidence:** 4

**Summary:**

The author proposes a curriculum dynamic graph invariant learning (CDGIL) framework that works for learning dynamic graph under distribution shift. They use extensive experiments to show that their proposed method out-performs all of the SOTA baselines that they considered.

**Strengths:**

The strength is the empirical evaluation looks promising.

**Weaknesses:**

The weakness are in many folds:

(1) Insufficient justification of motivation: This work suggests that they are the first to study the task of curriculum dynamic graph invariant learning. However this sounds like an A+B task to me and the author does not sufficiently justify why is this task so important. Also the reviewer is confused if the curriculum learning is a task or a method.

(2) Marginal contributions and novelty: This work gives weak justification on why this method is effective. Also it looks like a concatenation of existing methods. Also the theoretical supports are lacking.

(3) Overclaim of contributions: The author implies in their work that their method outperform ALL of the SOTA baselines on ALL of the dynamic graph datasets with distribution shifts. These statements look rather arrogant on an academic paper.

**Questions:**

Questions are raised in the weaknesses section.

---

### Meta-Review · Area_Chair_GfWr · 2023-12-06

**Metareview:**

This work introduces a CDGIL model to deal with the distribution shift problem in dynamic graphs. The reviewers have concerns about the soundness of motivation and limited novelty. The presentation should also be improved as pointed out by the reviewers. Since no rebuttal is provided to answer the questions, I opt for rejection.

**Justification For Why Not Higher Score:**

Please see the meta-review.

**Justification For Why Not Lower Score:**

N/A

---

### Decision · Program_Chairs · 2024-01-16

Reject